# Derivation of Kokumi γ-Glutamyl Peptides and Volatile Aroma Compounds from Fermented Cereal Processing By-Products for Reducing Bitterness of Plant-Based Ingredients

**DOI:** 10.3390/foods12234297

**Published:** 2023-11-28

**Authors:** Nabila Rodríguez Valerón, Tiffany Mak, Leonie J. Jahn, Juan Carlos Arboleya, Pia M. Sörensen

**Affiliations:** 1Basque Culinary Center, Facultad de Ciencias Gastronómicas, Mondragon Unibersitatea, 20009 Donostia-San Sebastián, Spain; nabila.rodriguez@alumni.bculinary.com; 2Harvard John A. Paulson School of Engineering and Applied Sciences, Harvard University, Cambridge, MA 02138, USA; 3Novo Nordisk Foundation Center for Biosustainability, Technical University of Denmark, Kgs, 2800 Lyngby, Denmarklejj@biosustain.dtu.dk (L.J.J.); 4BCC Innovation, Centro Tecnológico en Gastronomía, Basque Culinary Center, 20009 Donostia-San Sebastián, Spain

**Keywords:** kokumi, fermentation, upcycled miso, sustainability, aroma, side stream utilisation, bitterness

## Abstract

Current food production methods and consumption behaviours are unsustainable and contribute to environmental harm. One example is food waste—around 38% of food produced is wasted each year. Here, we show that two common food waste products, wheat bran and brewer’s spent grain, can successfully be upcycled via miso fermentation. During the fermentation process, kokumi γ-glutamyl peptides, known to increase mouthfulness, are produced; these include γ-ECG (oxidized), γ-EVG, γ-EV, γ-EE, γ-EF, and γ-EL. The profiles of kokumi peptides and volatile aroma compounds are correlated with koji substrate, pH, and enzymatic activity, offering straightforward parameters that can be manipulated to increase the abundance of kokumi peptides during the fermentation process. Correlation analysis demonstrates that some volatile aroma compounds, such as fatty acid ethyl esters, are correlated with kokumi peptide abundance and may be responsible for fatty, greasy, and buttery aromas. Consumer sensory analysis conveys that the bitter taste of vegetables, such as that in endives, can be dampened when miso extract containing kokumi peptides is added. This suggests that kokumi peptides, along with aroma volatile compounds, can enhance the overall flavour of plant-based products. This study opens new opportunities for cereal processing by-product upcycling via fermentation, ultimately having the potential to promote a plant-based diet.

## 1. Introduction

The planet is facing a global climate crisis. This is primarily attributed to unsustainable consumption and production patterns, of which the global food system is one of the major contributors [1,2]. Current industrialised meat production is a key driver of climate change [3], largely due to the high feed-to-food conversion ratios, resulting in systems that are resource intensive with low production efficiency compared to plant-based foods [4]. Another inefficiency of the current food system is its wastefulness, with 38% of food produced being wasted annually worldwide [5]. Only small amounts are currently being repurposed as animal feed or for other purposes [6]. Reducing meat consumption, shifting to more plant-based diets, and reducing food waste have, therefore, been proposed as key strategies for transitioning to more sustainable food systems [2].

To facilitate the necessary rapid transition in food consumption behaviours, taste is key [7]. However, one of the challenges with plant-based raw materials is that they often lack the umami, sweetness, mouthfeel, and aroma compounds characteristic of animal products [8]. “Green” or “beany” aromas such as hexanal, (E)-2-nonenal, and (E,E)-2,4-decadienal in plant-based ingredients present a challenge [9].

Another challenge with many plant ingredients, for example, with vegetables belonging to the Brassica and Cichorium genera, is that they have an inherent bitterness due to the presence of various bitter compounds such as glucosinolates [10,11], which are often less favoured by consumers [12].

Fermentation has been used to preserve food but also to improve the flavour and the bioavailability of nutrients from raw materials [13]. In particular, γ-glutamyl peptides have been shown to increase ‘mouthfeel’ amongst other desirable qualities and have been proposed as flavour enhancers in food [14]. Additionally, γ-glutamylization is being researched as a means of reducing bitterness by the synthesis of peptides such as γ-Glu-Phe, γ-Glu-Leu, or γ-Glu-Val from bitter amino acids like Phe, Leu, and Val through GGT (gamma-glutamyl transferase) catalysis [15]. These γ-glutamyl peptides cause a sensation called koku, a Japanese word describing the mouthfeels of thickness, richness, complexity, and continuity. These peptides have also been shown to be present in different fermented products like beer, cheese, soy sauce, sake, and miso [15,16]. Additionally, a concept called the “umamification” of plant-based materials suggests that fermentation can increase the concentration of amino acids and simple sugars to improve flavour [7], and kokumi peptides have been shown to increase umami [17], suggesting fermentation as a promising strategy for enhancing the favourable qualities of plant-based foods.

Previous studies have shown that kokumi peptides are present in miso prepared with alternative grains [16]. We were, therefore, interested in whether this concept can be expanded to additional substrates, especially cereal processing by-products, as a strategy to revalorise and reduce food waste. Wheat bran, a side product of milling wheat into white flour, and brewer’s spent grains (BSG), a by-product of beer filtration, are amongst the most common food production by-products; respectively, around 90 million tons and 32.5 tons are produced annually worldwide [6,18]. Combined with the high protein and other nutritional concentrations remaining after processing, both by-products are promising potential candidates for producing kokumi peptides [19,20] as well as improving the bioavailability of nutrients following fermentation [13].

In this study, we aimed to repurpose cereal processing by-products through miso fermentation as an approach to create a flavouring product that can be added to plant-based ingredients to enhance their desirability and favourable taste attributes. We particularly focussed on the generation of kokumi peptides through miso fermentation and investigated their relation to physicochemical properties and aroma compounds, as well as the potential to reduce bitterness—while increasing umami and mouthfulness—by performing sensory analysis. Existing studies on kokumi peptides have mainly focused on characterising human sensory receptor responses by reducing sugar and fat in food. In this study, we identified physicochemical parameters that could influence or even act as potential indicators for γ-glutamyl peptide formation.

## 2. Materials and Methods

### 2.1. Preparation of Koji and Miso

Miso is a Japanese seasoning usually prepared by fermenting steamed soybeans with salt and koji (*A. oryzae*). We prepared our upcycled misos as follows: Dried pearl barley (1 kg, Aurion, Denmark) was rinsed under cold water and then soaked in 4 L of tap water overnight at 7 °C [21]. Wheat bran (Kornby Mølle, Lynge, Denmark) and brewer’s spent grains (BSG) (commercial brewery, Copenhagen, Denmark) were mixed with distilled water at a 1:1 ratio. The koji and miso were prepared according to the method by Rodríguez Valerón et al. [16], with slight modifications. Two *A. oryzae* strains were used: S1, which is traditionally used for soybean miso (BF-3; Higuchi Matsunosuke Shoten Co., Osaka, Japan), and S2, which is traditionally used for barley koji (mugi senyo ki; Bio’c Co., Ltd., Uchida, Murocho, Toyohashi-shi, Aichi, Japan). Both strains were added in powdered form according to the manufacturer’s recommendations. Miso was prepared as shown in Table 1.

### 2.2. Physicochemical Parameters

To follow the miso evolution, physicochemical analyses were conducted after 0, 4, 8, and 12 weeks of fermentation. The physicochemical qualities of moisture, colour, pH, REDOX, and reducing sugar content were analysed according to previous studies [16,22,23]. All measurements were conducted in triplicate, except colour, which was measured 10 times.

#### 2.2.1. Colour Measurement

The colour was measured using a colour analyser CR400 (CR400, Konica Minolta, Inc., Tokyo, Japan) and analysed with SpectrMagic NX 3.1 software, using the CIE L*a*b* colour space, illuminant D65, and a 10° observer as a reference. The samples were measured following the method described in a previous study [16]. Processing of the miso in a Thermomix^®^ contributed to a uniform colour [16].

#### 2.2.2. Water Content, pH, and REDOX

Water content (%) was measured using a moisture analyser (XM-124-60 124/0.001 g, COBOS^®^, Barcelona, Spain) [16,24]. The pH was analysed on 10% (*w*/*v*) miso samples diluted in distilled water using a pH meter (Elite pH pocket testers, ThermoFisher Scientific, Waltham, MA, USA). The oxidation-reduction potential (REDOX) was measured by ORP/Redox tester (ORPTestr^®^ 10BNC, Oakton, Cole-Parmer, Cambs, UK) on a solution of 10% (*w*/*v*) miso samples diluted in distilled water [16].

#### 2.2.3. Reducing Sugar Content with Dinitrosalicylic Acid (DNS)

Reducing sugar content was measured with dinitrosalicylic acid (DNS) as previously described [16,22]. Miso samples (1% (*w*/*v*) were diluted in distilled water and filtered, and 1 mL of DNS solution was added. The samples were mixed, heated at 100 °C for 5 min, and cooled on ice [25]. The absorbance was recorded at 540 nm with a UV spectrophotometer (Cary 100 Series 2 Scan, Varian, Madrid, Spain) and analysed as previously described [16,17,18,19,20,21,22].

### 2.3. Protease Activity

Protease activity was measured as previously described [16]. Two grams of miso were mixed with 40 mL of distilled water and filtered. One mL of each miso solution was mixed with casein solution (0.6%) and kept at 37 °C for 10 min. Trichloroacetic acid (5 mL, 0.4 M) was added, and the samples were incubated for 20 min at 37 °C. The samples were then centrifuged (1147× *g*, 5 min), and 2 mL of the supernatant was mixed with 1 mL of Folin–Ciocalteu reagent, followed by addition of 5 mL of Na_2_CO_3_. After 30 min, absorbance was measured at 660 nm using a UV spectrophotometer (UH5300 Spectrophotometer, Hitachi, Ltd., Tokyo, Japan). Tyrosine was used for the standard curve, with one unit of protease activity corresponding to the amount of tyrosine produced from 1 mL of koji solution after one minute. All measurements were taken in triplicate.

### 2.4. Extraction and Identification of Kokumi Peptides by HPLC/Ms-Ms

Six different peptides were analysed based on suggestions from the literature: γ-Glu-Cys-Cly (analysed in its reduced and oxidized forms) (Sigma Aldrich, St. Louis, MO, USA), γ-Glu-Val-Gly (Trichem, Skanderborg, Denmark), γ-Glu-Val [26], γ-Glu-Glu, γ-Glu-Leu, and γ-Glu-Phe (Bachem, Frechen, Germany) [27]. Identification was carried out by targeted detection of peptides with standards. Quantifications are based on the chromatography areas [16].

The extraction method was based on our previous study [16]. Miso samples (800 mg) were combined with methanol (80% (*v*/*v*)), homogenised, and centrifuged (9000 rpm, 20 min). The liquid layer was collected (excluding the upper fat layer and the protein pellet), membrane-filtered, transferred to a chromatography vial, and kept at −20 °C until analysis [16].

Samples were analysed by LC-MS on a Vanquish LC coupled to an ID-X MS (ThermoFisher Scientific, Waltham, MA, USA), following a previous method [16]. Five µL of sample or standard was injected on a ZIC-pHILIC peek-coated column. The LC program is described in Rodriguez Valeron [16]. Data were acquired on the ID-X in switching polarities at 120,000 resolution, with an RF lens at 30%, normalized AGC target (Automatic Gain Control) at 25%, max IT at 50 ms, and *m*/*z* range of 120–500. For each peptide target, a targeted MS2 with isolation 1.6 *m*/*z*, HCD 20, 30,000 resolution in positive mode was also acquired (Appendix A). For each peptide, a unique MS2 fragment was selected for integration. The fragments were selected based on the pure standards MS2 spectra. These unique fragments, in addition to retention time differences, allowed for the differentiation between isobaric species [16].

### 2.5. Extraction and Quantification of Volatile Aroma Compounds by HS-SPME-GC/MS

The extraction method followed previously described studies [16,21]. Miso (20.0 g) was combined with ultrapure water (44.0 g), homogenised, and shaken at 25 °C for 2 h. The homogenate was filtered, and 4 mL were transferred to 10 mL gas-tight vials. Ten microliters of 2-methyl-3-heptanone was added to the homogenate as an internal standard [16,21]. The negative control was prepared with 20.0 g steamed barley, 4% (*w*/*w*) salt, 44.0 g distilled water, and 10.0 µL of internal standard. Solid-phase microextraction (SPME) fibre and polydimethylsiloxane/divinylbenzene (PDMS/DVB), in a quantity of 65 µm, was used to extract volatile compounds. Following extraction, the samples were incubated at 45 °C for 15 min, after which the SPME fibre was inserted into the headspace at 45 °C for 40 min and then inserted into the GC injection port. The GC-MS analysis was performed on a Thermo Scientific TRACE 1310 Gas Chromatograph equipped with a Thermo Scientific Q Exactive Orbitrap mass spectrometry system, as previously described [16]. The GC conditions followed our previously described method for alternative misos [16]. Retention index based on Thermo TG-5SILMS column using C7-C27 as external references. Concentration is expressed as 2-mehtyl-3-heptanone equivalent (µg/L). Method of identification: A, by comparison of the MS spectra with the NIST library; B, by comparison of RI (Kovat indices). All the compounds were identified by A and B methods. Data were acquired and analyzed with Thermo TraceFinder 4.1 software package [28].

### 2.6. Sensory Analysis

A kokumi-rich water-soluble extract was prepared from each miso following the water-soluble extract protocol (WSE) [29]. Each of the three miso samples (Table 1, 50 g) was mixed with filtered water at a ratio of 1:1 (*w*/*w*), autoclaved for 120 min at 40 kPa and 110 °C, and filtered through Whatman no. 2 filter paper (Whatman International Ltd., Maidstone, Kent, UK), with the aqueous phase obtained. The residue on the filter paper was re-extracted with water (42 mL) at room temperature and re-filtered. Both water extractions were combined and centrifuged at 9338× *g* for 15 min, and the liquid phase was reserved. Endive purée (prepared at a ratio of 2:1 endive/olive oil) was cooked at 180 °C for 20 min in the oven in vacuum bags and blended with a Thermomix^®^ (full speed, for 4 min). Extracts from each of the three miso samples were added to three independent samples of endive puree 20% (*w*/*w*). A control sample was prepared with endive pure and 20% (*w*/*w*) of tap water instead of miso extract.

For the consumer sensory analysis, each sample (30 g) was served at room temperature to a panel of 60 consumers in a room with controlled temperature and relative humidity (21 ± 2 °C; 35 ± 5% RH). Consumers tasted each sample and rated the intensity of bitterness, coating, aftertaste, and thickness using a 100-point general Labelled Magnitude Scale (gLMS) (0 = barely detectable, 100 = strongest imaginable) [30]. Aftertaste intensity was rated 20 s after placing the sample in the mouth, and thickness was rated after 5 s. Answers to the questionnaire were collected by RedJade Sensory Software version 2023 (RedJade Sensory Solutions, LLC, Martinez, CA, USA 2023). The samples were randomly coded with 3-digit numbers and offered to the consumers in random order. Consumers rinsed their mouths with water and unsalted crackers between samples.

The sensory analysis followed Regulation (EU) 20216/679 on the protection of personal data. The regulation and the experimental procedure were explained to the consumers, and consent for voluntary participation was given via consumer signature. The study was conducted in accordance with the Declaration of Helsinki, and the protocol was approved by the Ethics Committee of IEB-20220927-I.

### 2.7. Data Analysis

A one-way ANOVA test was conducted to find significant differences between samples taken at different times. A post hoc test was carried out using Tukey’s HSD. All data analysis were performed by the statistical package XLSTAT Version 2020.4.01 (Lumivero, Denver, CO, USA) [31]. Results were considered significant for *p* < 0.05. The correlation plot, heat map, and cluster analysis were produced with RStudio Desktop 2022.07.1+554 (RStudio, Boston, MA, USA) [32], with 95% confidence level and 0.001, 0.01, and 0.05 significance levels for the correlation analysis.

## 3. Results and Discussion

### 3.1. Miso Fermentation of Cereal Processing By-Products Generates γ-Glutamyl Peptides

Kokumi peptides have been shown to enhance the palatability and perceived mouthfulness of foods [33]. In a previous study, we demonstrated that the miso fermentation of regional cereal grains results in the generation of γ-glutamyl peptides [16]. We were, therefore, interested in whether the miso fermentation of cereal byproducts is able to produce kokumi peptides as well. We selected two main cereal production and processing by-products, wheat bran and brewer’s spent grain (BSG), and introduced them at different stages of the miso fermentation. Wheat bran was used as the key substrate for the preparation of koji in this study based on evidence of wheat bran being utilised in traditional and industrial koji-making practices [34,35,36,37], and BSG was used as the miso substrate based on its high residual protein and nutrient contents that are comparable to other common miso substrates [6,20]. We also included barley, a substrate that was demonstrated to produce γ-glutamyl peptides through miso fermentation [16], and tested different combinations of miso fermentation as detailed in Table 1. To investigate the potential effect of the genetic influence of the koji fungus, *A. oryzae*, on the fermentation process, we also tested different strains of *A. oryzae* (S1 and S2) for specific combinations (Table 1).

A previous study demonstrated that all but one of the γ-glutamyl peptides that were identified in grain-based miso fermentation [16] were also present in the miso samples prepared using cereal-processing by-products, including γ-ECG (oxidized), γ-EE, γ-EL, γ-EF, γ-EV, and γ-EVG. Likewise, the reduced form of γ-ECG was not detected, most likely due to natural oxidation during the fermentation process (Figure 1) [16].

Cluster analysis of the different kokumi peptides and miso shows that the miso made with wheat bran koji is clustered into one group, and the one made with barley koji is clustered into another (Figure 1A). This observation demonstrates that the koji substrate is an important parameter for the abundance and identity of kokumi peptides in subsequent miso fermentations, as suggested previously [16].

Looking more closely at the individual γ-glutamyl peptides identified, γ-ECG oxidized was present in all of the samples after 12 weeks of fermentation (Figure 1A). Notably, the abundance of γ-ECG is inverse to the abundance of γ-EVG and γ-EV in the different samples (Figure 1A). This trend might be explained by the mechanism suggested by Sofyanovich et al. [38], where γ-ECG is the precursor of γ-EV and γ-EVG. The synthesis of γ-EV and γ-EVG is mediated through the transfer of the γ-glutamyl residue from γ-ECG to valine via the GGT (glutamyl transferase) pathway or the dipeptide Val-Gly (VG), respectively. The abundance of γ-EVG is promising because it is the most potent kokumi peptide found and studied in a wide range of fermented products [33].

Interestingly, γ-EE was identified in all samples at the end of the fermentation process (Figure 1A). In our previous study on alternative miso, this peptide was identified in barley koji, so this confirms the relation between koji substrate and the generation of kokumi peptides [16]. The mechanism of γ-glutamyl dipeptides is led by γ-glutamyl transfer reaction via GCL and GCS. Thus, free amino acids such as glutamic acid are a limiting factor to synthesise γ-EE [39].

Peptide γ-EL was found in all miso samples, with higher abundance observed in the two misos that were made using BSG as the miso substrate (Figure 1A). This could be explained by the higher concentration of free leucine in BSG (6.1 *w*/*w* %) than in barley (0.3 *w*/*w* %) [40]. This peptide has also been reported to be present in various fermented foods, including soy sauce, sourdough, parma dry-cured hams, gouda, and parmesan cheese, as well as occurring naturally in various edible beans [33].

By contrast, γ-EF was only detected in wheat bran S1/BSG in the last stage of the miso fermentation (Figure 1D). The abundance of this dipeptide is slightly decreased during the fermentation time in the remaining samples until it reaches zero (Figure 1B–D). It has been found in various fermented products, such as miso and soy sauce [33].

### 3.2. Physicochemical Changes in Fermentation Processes Influence the Generation of γ-Glutamyl Peptides

It is evident from this study, as well as previous studies, that components in the raw materials used to prepare koji contribute to the formation of precursors such as γ-ECG and the release of different amino acids [16]. Our previous results have also shown that physicochemical changes occurring during the fermentation process, including pH, REDOX, reducing sugar, and colour, are factors that are correlated with differential kokumi peptide formation [16]. Since physicochemical changes can both influence and be influenced by related enzymatic activities during the fermentation process, monitoring the physical and chemical parameters (Appendix A) not only provides a better understanding of the production of kokumi γ-glutamyl peptides but can also become a useful tool for tracking the progress of fermentation, especially for unconventional substrates like by-products.

In this study, we measured the changes in pH levels, protease activity, reducing sugar, water content, and colour, across all samples throughout the 12-week fermentation period, taking samples every 4 weeks. During the fermentation process, the pH levels of three of the four misos (wheat bran S1/barley, wheat bran S2/barley, and barley S1/BSG) followed a similar pattern, whereby the pH level initially decreased from 6 ± 0.06 to 4 ± 0.06 in the first eight weeks, then increased slightly back to around 6 in the final stage of the fermentation (Figure 2A). This pattern of modulation in pH profile is consistent with what we observed in our previous studies of miso fermentation [16,40] In contrast, the pH level of the miso made from wheat bran S1/BSG showed an unusual pH profile, where a gradual increase from 6.63 ± 0.06 to 9.43 ± 0.06 (Figure 2B) was observed over the 12 weeks, without the pH ever reaching below 6, indicating an unusual fermentation process. This unexpected change in pH seems to have an impact on the abundance of the γ-EV peptide in particular (Figure 2B). This pattern and difference suggest that the combination of substrates impacts the pH behaviour and, consequently, kokumi peptide generation. In addition to being a good indicator for monitoring the fermentation process, pH changes may also serve a functional role in the synthesis of γ-dipeptides such as γ-EE, γ-EF, and γ-EL as it influences the ionic form of amino acids, like glutamic acid to glutamate. Glutamate is also an important amino acid for the γ-glutamylation reaction and has been implicated in dampening bitterness from phenylalanine, leucine, and valine [41], a taste attribute that we will further explore in Section 3.4.

We observe in this study that pH levels are strongly correlated with the changes in enzymatic activities during fermentation (*p* < 0.01), as shown in Appendix A. The results show the lowest activity for barley S1/BSG (0.80 ± 0.01 U/g), followed by wheat bran S2/barley (1.09 ± 0.02 U/g), wheat bran S1/BSG (1.23 ± 0.01 U/g), and the highest for wheat bran S1/barley (1.70 ± 0.02 U/g) (Appendix A) for the final time point. It was also found that the koji and miso substrates are not the only determining factors for the fermentation process. Based on results from the two misos that used the same koji and miso substrates (wheat bran S1/barley and wheat bran S2/barley), we observed very different enzymatic activities (Figure 2C). This suggests that the *A. oryzae* strain also has an influence on the enzymatic activity during fermentation. According to a study by Lin et al. in [42], protease activity reaches its highest point at pH levels ranging from 6 to 7, depending on the specific *A.* strain. Meanwhile, the synthesis of γ-glutamyl peptides involves three distinct enzymes, including GGT (γ-glutamyl transferase), glutaminase, and either GCS (γ-glutamylcysteine synthetase) or GCL (glutamate-cysteine ligase). In vitro, these enzymes catalyse the creation of γ-glutamyl dipeptides from glutamate and amino acids with an optimal pH above 7.5 [43]. These observations suggest that the strain of *A. oryzae* used could also have an influence on γ-glutamyl peptide formation and remains to be further explored.

Another physicochemical property we investigated were the parameters in relation to colour (a*, b*, and L*). Our previous study showed that a* and b* were also correlated with kokumi peptides, particularly for γ-EL and γ-EV [16]. According to a study by Wang et al. in [44], the impact of colour on consumer preferences in product appearance was examined, where on average, high a* (reddish) was associated with higher miso quality, whereas b* (blueish) and L* (luminosity) showed the opposite effect. Luminosity (L*) affects the darkness/reddish characteristic of miso and is closely related to its palatability and acceptance. The study described in this paper found that L* was negatively correlated with γ-EE, γ-EF (*p* < 0.05), and γ-EL (*p* < 0.01) (Appendix A). It was also observed that L* is negatively correlated with pH (Appendix A), and there is an especially pronounced decrease in L* accompanying the increase in pH for the wheat bran S1/BSG miso (Figure 2D). Based on our earlier conclusion of pH as a potential indicator for fermentation progress (Figure 2D), as well as subsequent characterisations of volatile aroma profile in Section 3.3, this suggests that luminosity could serve as a potential indicator for predicting the desirable characteristics for miso-based fermentation.

### 3.3. Volatile Aroma Compound Profiles of Upcycled Miso Reveal Correlation with γ-Glutamyl Peptides

Aroma plays an important role in food acceptance. Some interactions of plant-based protein and aromas have been studied, suggesting that compounds such as hexanal, (E)-2-nonenal, and (E,E)-2,4-decadienal can lead to consumer rejection. However, fermentation is able to reduce these compounds, making plant-based ingredients more palatable [9].

To explore this further, we conducted an HS-SPME-GC/MS analysis and found a total of 125 volatile organic compounds across the different miso samples (Appendix A). We first performed a correlation analysis between the kokumi peptides and volatile aroma compounds and identified 67 compounds that showed correlative association with the γ-glutamyl peptides (Appendix A) and 63 in particular that were present in the last stage of the miso fermentation (Figure 3). These compounds included three acids, four alcohols, six aldehydes, thirty-one esters, ten ketones, five phenols, and four pyrazines. These results corresponded with our previous study on alternative grain-based miso, where the most represented volatile aroma group present was esters [16].

Cluster analysis of the volatile aroma compound profiles revealed that three of the misos (wheat bran S1/barley, wheat bran S2/barley, and barley S1/BSG) belong to the same cluster, whereas the miso made of wheat bran S1/BSG belongs to a separate second cluster (Figure 3). This observation may be related to the unexpected differences in physicochemical changes that were exhibited during the fermentation process, especially the non-acidic pH levels for the wheat bran S1/BSG miso (Figure 2B), which could serve as a potential explanation for the unusual aromatic attributes of this sample, a point that we will return to later in this section.

First, focusing our analysis on the volatile aroma profile of the main clusters, it was observed that the esters were amongst the group of volatile compounds most strongly associated with kokumi peptides, especially high-fatty acid ethyl esters like hexadecanoic acid methyl ester, hexadecanoic acid ethyl ester, 9,12-octadecadienoic acid (Z,Z)-methyl ester, elaidic acid methyl ester, linoleic acid ethyl ester, and ethyl oleate (Figure 3). This observation is in line with our previous study on alternative grain-based miso, where we also observed high concentrations of ester aroma compounds due to the degradation of fatty acids [16]. These molecules are known to have waxy, fatty, oily, and fruity aromas. Most of them are produced during long-term fermentation processes lasting 3–6 months [45,46]. They might be involved in the relationship between the koku sensation and fat, contributing to a mouthfeel of coating and lingering as well as an aftertaste, as has also been suggested in a previous study that showed that kokumi peptides can enhance the fatty taste [26].

Compounds that cause fruity aromas are also present prominently in the main cluster; these include compounds from the ester group, such as isobutyl acetate, 2-methyl-butanoic acid- ethyl ester, 3-methyl-butanoic acid- ethyl ester and isoamyl lactate, as well as phenylethyl alcohol (Figure 3); the latter has been associated with a rose-honey-like aroma [44]. Phenylethyl alcohol is particularly noteworthy, as it is found to be present in all three misos in the cluster (Figure 3) and has been reported to be produced by degrading the amino acid phenylalanine [46], which could be a possible explanation for why these misos do not contain γ-EF (Figure 1A).

Within the main cluster, the three misos can be further sub-clustered into two different groups based on their volatile aroma profiles. Interestingly, wheat bran S1/barley and wheat bran S2/barley did not cluster into the same group, suggesting that the strain of *A. oryzae* also has a notable influence on the volatile aroma profile of the final miso product, which may be influenced by the differences in inherent enzymatic activities of the strain of fungi, as observed in Figure 2C. Nonetheless, we still observe a few aromatic compounds in common, including 2-methoxy-4-vinylphenol, which is often found in soy sauce and is associated with a burnt scent [47], and trimethyl–pyrazine, which is associated with roasted and chocolate aromas [48]. These compounds can likely be attributed to the presence of lignin in wheat bran and provide evidence for koji and miso substrates still contributing towards the final aromatic profile of the fermented miso.

In the miso made from wheat bran S1/barley, we also detected other aroma compounds that are found in many fermented products, including red sufu, cacao beans, and soybean products, and are known for their sweet, fruity, and floral notes [49]. In particular, benzaldehyde and α-ethylidene are formed through the degradation of phenylalanine [49], and menthol creates a menthol aroma [50].

In terms of the other cluster, the miso made up of wheat bran S1/BSG exhibited a high concentration of phenol compounds, including 2,3-dimethyl, which is not typically associated with miso flavour. This phenomenon may be attributed to the breakdown of the high concentration of lignin in cereal bran and BSG, producing phenol compounds [44]. Other phenolic compounds detected include 4-ethyl-2-methoxy, which imparts smoky and spicy notes [51], and 2-phenylpropenal, a compound responsible for the green, honey, alcoholic, sweet, caramel, bread, and coffee aromas found in fermented rice bran [45]. Compounds belonging to the pyrazine group, such as trimethyl–pyrazine, which is associated with a roasted aroma [49], were detected in higher concentrations than in the other two misos made from wheat bran koji (Figure 3). This high concentration of pyrazine compounds may be attributed to the high pH during the fermentation of this miso, which could accelerate the formation of pyrazines through the Maillard reaction between reducing sugars and amino acids [52].

Overall, we observe a strong negative correlation between volatile aroma compounds and γ-glutamyl peptides (Figure 4). This observation may be explained by the cabalism of amino acids; for instance, aldehydes and alcohols are the primary compounds produced through the Ehrlich pathway by various fungal enzymes during fermentation [21] and Strecker degradation [53]. Therefore, the synthesis of kokumi peptides may be hindered as a result of the degradation of amino acids.

Volatile aroma compounds and kokumi peptides may intensify the experience of the koku sensation, a Japanese term used to describe the presence of kokumi peptides. The link between high levels of fatty acid ethyl esters and kokumi peptides could be a possible explanation for these characteristic associations with the koku sensation, such as aftertaste, lingering, and coating, and remains a question to be further explored.

### 3.4. Addition of Kokumi Peptides Reduces Bitter Taste

Bitterness is not a preferred taste for many people. Many nutritious foods like vegetables and cereals contain compounds such as isothiocyanates or glucosinolates that can cause bitterness [54]. These flavours can be further accentuated in plant-based upcycled products [55], preventing wider acceptance and changes in dietary habits towards more planetary friendly food options [8].

Kokumi peptides have been studied to increase umami, saltiness, sweetness, and fatty texture [17]. In particular, bitterness has been implicated to be reduced by fermentation through γ-glutamylation reactions [39]. Based on this observation, our study tested whether the γ-glutamyl peptides produced during the fermentation of our upcycled miso were able to reduce bitterness when added to plant-based ingredients. Endive (*Cichorium intybus var. foliosum*) was selected as the vegetable for our sensory analysis due to its distinct bitter taste produced by glucosinolates [12]. Using a method of water extraction, we obtained an extract from each of our upcycled misos that was rich in kokumi peptides, and after adding them to the endive purée in the same concentrations, performed sensory analysis for each of the samples (for details, see Section 2). It should be noted that extraction and sensory analysis were only performed for three of the four misos (barley S1/BSG, wheat bran S1/barley, and wheat bran S2/barley). The wheat bran S1/BSG miso was excluded due to the undesirable aroma profile and unexpectedly high pH observed throughout the fermentation. For the sensory analysis, an intensity scale was used to evaluate the bitterness, aftertaste, coating, and thickness. The questionnaire also included questions about the participants’ attitudes to vegetables in order to establish whether their preferences and previous experiences might affect their perception of bitterness.

Results from the sensory analysis confirmed our hypothesis that the addition of kokumi-rich miso extracts could reduce the perceived bitterness in the endive samples; we observed that this was the case for two of the three samples tested. The endive sample containing extract from the barley S1/BSG miso was perceived to be the least bitter, followed by wheat bran S1/barley, control (water instead of extract added), and wheat bran S2/barley (Figure 5). This result may partly be explained by the total abundance of γ-glutamyl peptides in each sample (Figure 5B), where barley S1/BSG miso has the highest amount, followed by wheat bran S2/barley and wheat bran S1/barley. Interestingly, despite wheat bran S1/barley having a lower abundance of γ-glutamyl peptides overall, it shows a higher absolute quantity of γ-EVG than wheat bran S2/BSG (Figure 5B). γ-EVG has been reported as a potent kokumi peptide with a sensory activity that is 12.8-fold stronger than GSH [56]. The higher abundance of γ-EVG in the wheat bran S2/BSG could, therefore, be a possible explanation for why its addition to endive purée results in lower perceptions of bitterness than the wheat bran S1/BSG sample.

In terms of the other attributes assessed in the sensory analysis, aftertaste was the only attribute other than bitterness that significantly differed between samples (Table 2). This could be the result of the lingering bitterness, especially for the sample containing the extract from the wheat bran S2/barley, which was perceived to be the most bitter and was ranked the highest for aftertaste (Figure 5A). Other studies have shown that the extent and impact between the bitter taste and aftertaste may be related to the regional distribution of bitter taste receptors on our tongue [57]. Future experiments should investigate whether γ-glutamyl peptides have differential effects on different bitter taste receptors.

Finally, our study found that participants who reported liking vegetables perceived the bitterness as being lower than those who did not. Furthermore, the perceived reduction in bitterness was also greater amongst those participants who reported eating vegetables more often (Appendix A). These observations are in agreement with the study by Vecchio et al. [12], where positive preferences toward bitterness were correlated with greater compensatory health beliefs. This opens up a promising area of research for influencing consumer preferences towards more plant-based diet options by reducing the less desirable flavour attributes of plant-based ingredients. 

## 4. Conclusions

In conclusion, this study demonstrates the possibility of producing γ-glutamyl peptides through miso fermentation of two common cereal processing by-products, wheat bran and BSG. Overall, it was observed that the choice of koji and miso substrates played a crucial role in the total abundance of kokumi peptides at the end of the 12-week fermentation, with the combination of using barley as the koji substrate and BSG as the miso substrate generating the greatest absolute concentration of peptides overall. Physicochemical characterisation during the fermentation process revealed that the optimum pH for the production of the peptides is between 5 and 6. Analysis of the volatile aroma profiles of the respective misos demonstrated that particular volatile compounds, specifically high fatty acid ethyl esters, are correlated to the abundance of kokumi peptides. Given these observations, the choice of fermentation substrates, as well as monitoring physicochemical changes like pH, are suggested to be important factors for creating favourable conditions to generate more desirable flavour attributes like kokumi peptides and the volatile aroma compounds when fermenting plant-based ingredients. Furthermore, we showed that extracts from the misos that were rich in kokumi peptides reduced the bitter taste in vegetables. To our knowledge, this is the first time this has been demonstrated. These results show great potential for revalorising food production by-products through fermentation by creating flavouring products that can enhance the desirability and favourable taste attributes of plant-based ingredients. Such strategies can contribute towards shifting current food consumption patterns and transforming our food production systems to more planet friendly, healthy, and sustainable futures.

## Figures and Tables

**Figure 1 foods-12-04297-f001:**
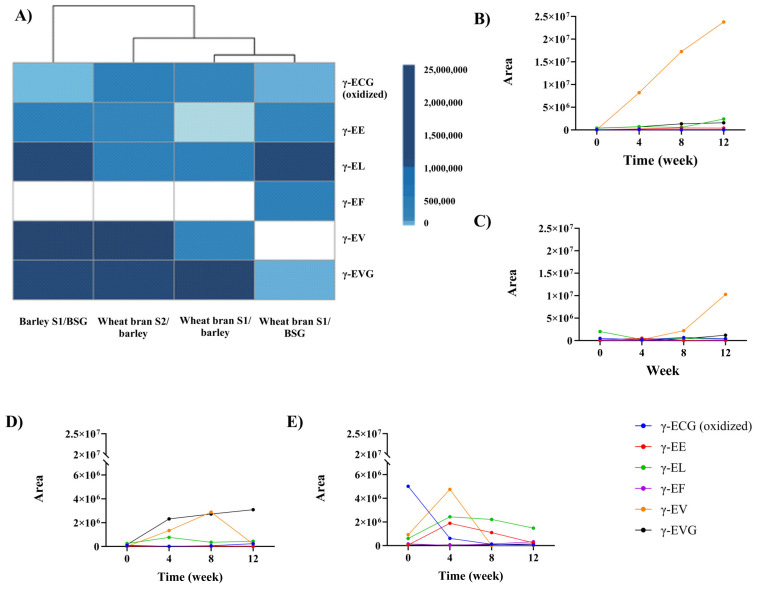
(**A**) γ-Glutamyl peptides detected in barley, BSG, and wheat bran miso after 12 weeks of fermentation, including cluster analysis for samples. The abundance of γ-glutamyl peptides is based on chromatography areas, which are unitless. Relative abundance can be compared for the same peptide but not across different peptides. (**B**–**E**) Evolution of γ-glutamyl peptides over time: (**B**) barley S1/BSG, (**C**) wheat bran S2/barley, (**D**) wheat bran S1/barley, (**E**) wheat bran S1/BSG. Appendix A shows MS information.

**Figure 2 foods-12-04297-f002:**
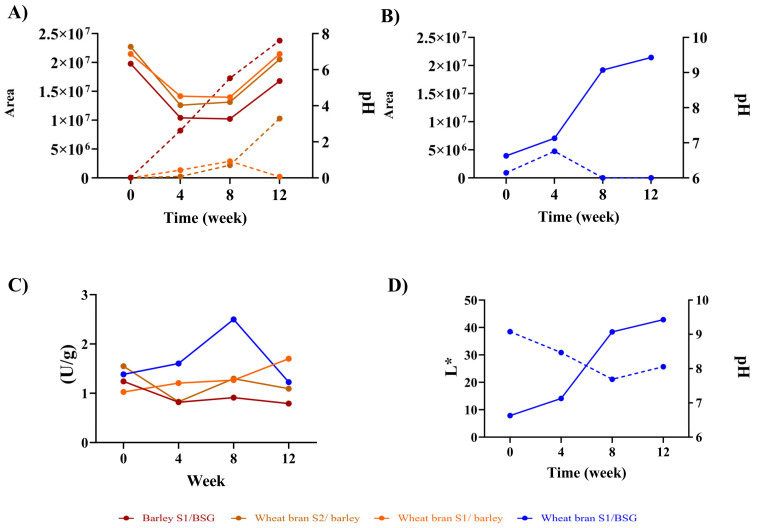
Physicochemical characteristics during the miso fermentation. (**A**) pH and γ-EV for the first cluster, (**B**) pH and γ-EV for the second cluster, (**C**) proteases activity (U/g), (**D**) L* luminosity vs. pH. One-way ANOVA and post hoc analysis (Tukey test) with 95% confidence levels are shown in Appendix A. All parameters were measured in triplicate, excluding colour, where *n* = 10. Legend: pH: dotted line, peptides: continuous line.

**Figure 3 foods-12-04297-f003:**
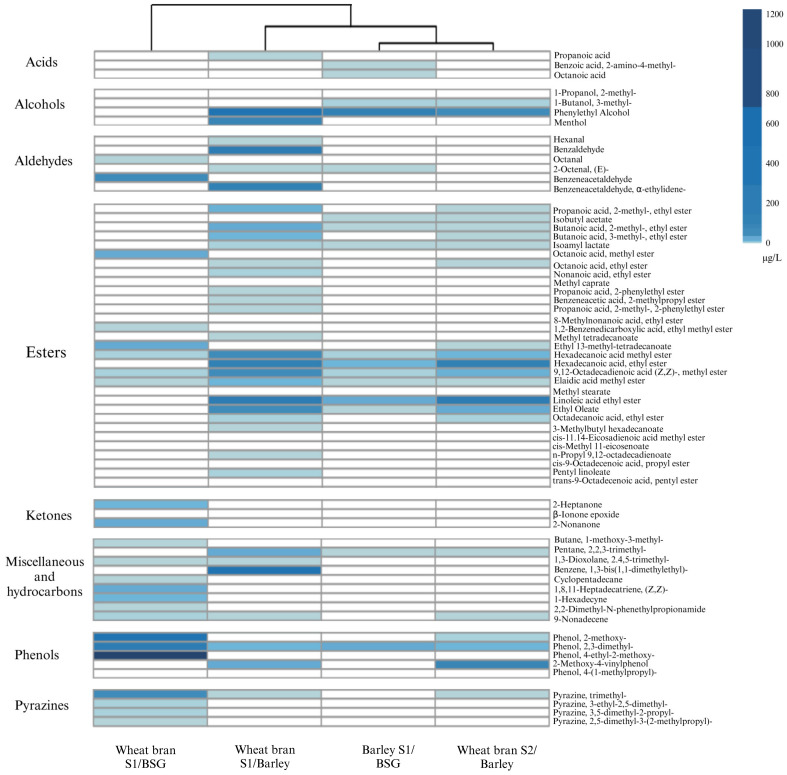
Volatile aroma compounds were identified in the miso after 12 weeks of fermentation. All the volatile compounds shown have a significant correlation with the kokumi peptides (Pearson’s coefficient, 95% confidence level). Kokumi peptides correlation plot and statistical analysis are available in Appendix A, respectively. Compounds in bold are present in the highest concentrations and are mentioned in the manuscript.

**Figure 4 foods-12-04297-f004:**
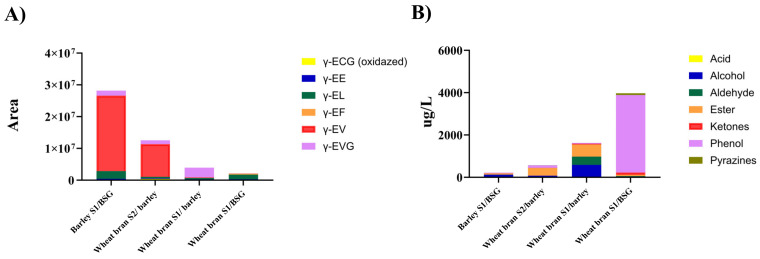
(**A**) Total abundance of γ-glutamyl peptides in each sample after 12 weeks of fermentation as based on chromatography areas. (**B**) Total concentrations of volatile aroma compounds in each sample after 12 weeks of fermentation.

**Figure 5 foods-12-04297-f005:**
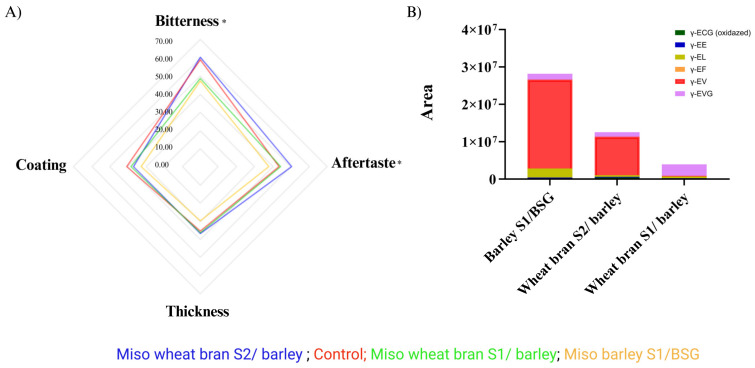
(**A**) Results from the consumer study: Intensity perception of aftertaste, coating, thickness, and bitterness when a kokumi-rich extract from each of the misos was added to endive purée. Legend: Red: control; Orange: barley S1/BSG; Blue: wheat bran S2/barley; Green: wheat bran S1/barley. Attributes followed by * means significant differences (*p* < 0.05). (**B**) Sum of γ-glutamyl peptides at 12 weeks.

**Table 1 foods-12-04297-t001:** Ingredients for miso preparation in terms of koji substrate, miso substrate, the strain of *A. oryzae.* and sample codce. The combinations of substrate and koji were based on a previous study [16].

Sample Code	Koji Substrate	Miso Substrate	*A. oryzae* Strain
Barley S1/BSG	Barley	BSG	S1
Wheat bran S2/barley	Wheat bran	Barley	S2
Wheat bran S1/barley	Wheat bran	Barley	S1
Wheat bran S1/BSG	Wheat bran	BSG	S1

**Table 2 foods-12-04297-t002:** Results of one-way ANOVA and post hoc analysis (Tukey HSD) of intensity perception in a gLMS for bitterness, aftertaste, thickness, and coating. Samples within the same column followed by different letters were significantly different; bold font indicates significant *p*-values (*p* < 0.05).

Sample	Bitterness Intensity	Aftertaste Intensity	Thickness Intensity	Coating Intensity
Control	59.00 ab	43.350 ab	35.52 a	40.45 a
Miso Barley S1/BSG	47.10 c	37.87 b	29.95 a	32.72 a
Miso wheat bran S2/barley	60.45 a	50.62 a	36.53 a	36.93 a
Miso wheat bran S1/barley	48.75 bc	44.27 ab	36.07 a	38.22 a
*p*-value	**0.003**	**0.042**	0.339	0.360

## Data Availability

Data is contained within the article and Appendix A.

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
