# Peer review of "Derivation of Kokumi γ-Glutamyl Peptides and Volatile Aroma Compounds from Fermented Cereal Processing By-Products for Reducing Bitterness of Plant-Based Ingredients"

_foods, 2023, doi:10.3390/foods12234297_

Round 1
Reviewer 1 Report
Comments and Suggestions for Authors
Paper is well written and presents interesting possibility of using cereal processing by-products for reducing bitterness. Must be improved chromatographic part.
Some issues I found: 1. In all manuscript names of chemicals please write not with capital letter, see e.g. L. 169, 170, 163 as many other; 2.Because of Add table with S1 (with MS2 precursors) to main manuscript. It will be more useful for further readers; 3. Describe what was " AGC"; 4. What was a mass of 2-mehtyl-3-heptanone added to the sample, not only volume and concentration. (0,27 ug ai I properly calculated); 5. Instead of Cyclohexanol, 5-methyl-2-(1-methylethyl)-, (1α,2α,5α)- please add name menthol; 6. 3,5-di-tert-Butyl-4-hydroxybenzaldehyde is a plasticiser, please remove from Table or mark it; 7. Methyl α-chlorophenylacetateis an artefact, please remove; 8. Benzoic acid, 2-amino-4-methyl- can't be at this time in comparison to other compounds. Please re-check; 9. Benzeneacetaldehyde, α-ethylidene could be an artefact. Please re-check; 10. Dimethyl phthalate is plasticiser, please re-check data; 11. Dibutyl phthalate is plasticiser, please re-check data; 12. Z)-Oxacyclopentadec-6-en-2-one could be an artefact. Please re-check; 13. re-check proper identification of Pentyl linoleate; 14. re-check proper identification Octan-2-yl palmitate 15. 1,3-Benzenedicarboxylic acid, bis(2-ethylhexyl) ester could be artefact. Please re-check; 16. 2,5-di-tert-Butyl-1,4-benzoquinone could be plasticiser, please re-check data; 17. Cyclopentadecane is artefact. Please re-check; 18. Phenol, 2-(1,1-dimethylethyl)-4-methyl- is an artefact, please re-check; 19. 13-Oxabicyclo[10.1.0]tridecane is an artefact, please re-check;Author Response
Dear reviewer
Thank you for your valuable feedback. We have addressed all of your comments and suggestions in the manuscript.
Reviewer 1
Paper is well written and presents interesting possibility of using cereal processing by-products  for reducing bitterness. Must be improved chromatographic part. 
Some issues I found:
- In all manuscript names of chemicals please write not with capital letter, see e.g. L. 169, 170, 163 as many other;
Thank you. This has been addressed throughout the manuscript.
- Because of  Add table with S1 (with MS2 precursors) to main manuscript. It will be more useful for further readers;
Thank you for this comment. We appreciate your input and agree that the table included in the manuscript would be beneficial for readers. However, we have decided to keep the supplemental material separate as it may be too detailed for readers who are not familiar with the field. Our aim is to cater to a broader audience, including non-experts. In case anyone is interested in delving into the details, we have provided them in the supplemental materials.
- Describe what was " AGC":
Thank you, the definition has been added to the methods section.
- What was a mass of 2-mehtyl-3-heptanone added to the sample, not only volume and concentration. (0,27 ug ai I properly calculated);
We are unsure if we understand your request correctly. We give both volume (10uL) and concentration (27.2 mg/L) in the methods section, and the mass can very easily be calculated from this (it is 0.027ug). Also, since the compound comes in liquid form, the volume is more relevant for the experimenter who wishes to repeat the experiment. Please let us know if we misunderstood your request.
- Instead of Cyclohexanol, 5-methyl-2-(1-methylethyl)-, (1α,2α,5α)- please add name menthol;
Thank you. This has been changed in figure 3 and in the text.
- 3,5-di-tert-Butyl-4-hydroxybenzaldehyde is a plasticiser, please remove from Table or mark it;
Thank you. This compound has been removed.
7. Methyl α-chlorophenylacetateis an artefact, please remove;
This compound has been removed. Thank you for the feedback.
8. Benzoic acid, 2-amino-4-methyl- can't be at this time in comparison to other compounds. Please re-check;
Due to the fact that this compound is not relevant in miso, it has been removed.
- Benzeneacetaldehyde, α-ethylidene could be an artefact. Please re-check;
Thank you for the feedback. This compound is naturally found in Phallus impudicus, Theobroma cacao.
https://pubchem.ncbi.nlm.nih.gov/compound/Benzeneacetaldehyde_-alpha-ethylidene#section=Structures
10. Dimethyl phthalate is plasticiser, please re-check data;
Thank you for this commentAccording to a reference that compound could be part of the agricultural process:
That compound has been studied on germination and antioxidant properties in different crops in China. Due to the fact that we do not have the route of the substrate, we suggested that the compound came from the grains and upcycled products. In general, that compound is reduced at the end of the fermentation. Please see thislink :
https://link.springer.com/article/10.1007/s11356-015-5855-y
11. Dibutyl phthalate is plasticiser, please re-check data;
Dibutyl phthalate (DBP) is the most widely used plasticizer for agricultural mulching films and one of black soil's most common organic pollutants. The same explanation applies to dimethyl phthalate regardingthe origin of the grains. It is very interesting that it is reduced through the fermentation processes. Please see the link below for more information: 
https://www.sciencedirect.com/science/article/abs/pii/S0045653522000431
12. (Z)-Oxacyclopentadec-6-en-2-one could be an artefact. Please re-check;
Thank you for the comment. This compound has been removed since it could be a possible contamination.
- re-check proper identification of Pentyl linoleate;
Thank you for this comment. We have rechecked this compound.
- re-check proper identification Octan-2-yl palmitate
Thank you for this comment. We have rechecked this compound.
15. 1,3-Benzenedicarboxylic acid, bis(2-ethylhexyl) ester could be artefact. Please re-check;
Thank you for the comment. We have removed this compound due to it possibly being a contamination.
- 2,5-di-tert-Butyl-1,4-benzoquinone could be plasticiser, please re-check data;
Thank you for this comment,, We have checked this compound. The compound concentration is 0 at the end of the fermentation. However, it has been removed.
17. Cyclopentadecane is artefact. Please re-check;
Thank you for this comment. We have deleted this compound.
- Phenol, 2-(1,1-dimethylethyl)-4-methyl- is an artefact, please re-check;
Thank you for the comment; we have checked it. The compound concentration is 0 at the end of the fermentation. It is considered a food additive in countries such as Canada.
https://www.canada.ca/en/health-canada/services/chemical-substances/challenge/batch-8/bha.html
- 13-Oxabicyclo[10.1.0]tridecane is an artefact, please re-check;
Thank you for the comment, it has been removed due to possible contamination in one sample.
Reviewer 2 Report
Comments and Suggestions for Authors
Manuscript Title: Derivation of kokumi g-glutamyl peptides and volatile aroma gDerivation of kokumi compounds from fermented cereal processing by-products for reducing bitterness of plant-based ingredients. In this manuscript significant work was done however, there are few minor issues with the manuscript which needs to be resolved before considering it for publication.
Abstract section: Please include the numeric values in this section.
Introduction: Rewrite this part. Try to explain the scientific reasons which insists the authors to perform the present investigation.
In this study, we aimed to repurpose cereal processing by-products through miso fermentation as an approach to create a flavouring product that can be added to plant-based ingredients to enhance their desirability and favourable taste attributes. We particularly focussed on the generation of kokumi peptides through miso fermentation and investigated their relation to physicochemical properties and aroma compounds, as well as the potential to reduce bitterness – while increasing umami and mouthfulness – by performing sensory analysis. The intention of the study is to propose new strategies through exploring the role of taste to enable the necessary transitions towards more sustainable future food systems and diets.
Section 2.2. Preparation of koji and miso
Add information related to media composition which was used to sustain the growth of microbial strain used during this study.
Comments on the Quality of English Language
Rather than write we did this, we observed this, we found that, represent the observations scientifically.
Author Response
Dear reviewer
Thank you for your valuable feedback. We have addressed all of your comments and suggestions in the manuscript.
Manuscript Title: Derivation of kokumi g-glutamyl peptides and volatile aroma gDerivation of kokumi compounds from fermented cereal processing by-products for reducing bitterness of plant-based ingredients. In this manuscript significant work was done however, there are few minor issues with the manuscript which needs to be resolved before considering it for publication.
Abstract section: Please include the numeric values in this section.
Thank you for the comment, we hesitate to include the numeric values since they don’t seem relevant for an abstract. We will adjust according to yours and the editor’s recommendation if necessary.
Introduction: Rewrite this part. Try to explain the scientific reasons which insists the authors to perform the present investigation.
Thank you for this comment. We have addressed this in the manuscript as follows.
Section 2.2. Preparation of koji and miso
Add information related to media composition which was used to sustain the growth of microbial strain used during this study.
Thank you for this comment. We have added this information to the manuscript. The microbial strain was acquired as a powder, and as is usually the case in koji making, the microbe was added in powdered form to the rice in the miso-making process. No media was involved in sustaining the microbe.
Rather than write we did this, we observed this, we found that, represent the observations scientifically.
Thank you. Like many other science writers, we are partial to using the active rather than passive voice. Nevertheless, we have taken your advice and addressed this throughout the document.
Reviewer 3 Report
Comments and Suggestions for Authors
Dear Authors,
In general, the manuscript is good to read, the structure of the work is clear. After reading the paper, it is clear that the authors performed a lot of experiments and analyzes in order to obtain detailed research results. I present my comments below:
1. A minor mistake. Now is: "2.2. Preparation of koji and miso" should be "2.1. Preparation of koji and miso".
2. The authors should explain what miso paste is in the methodology. Not every reader has encountered this regional product.
3. Chapter 2.5. Please comment it. Why did the authors choose this particular fiber and this particular chromatographic column for their research?
4. Lines 253 - 263. Starting with: "We selected two main cereal production..." and ending with: "... of miso fermentation as detailed in Table 1". This is information that relates to the methodology and is unnecessarily in this chapter.
5. In graphs B, C, D, E in Figure 1, the authors could add standard deviations if the graphs are based on average scores. Similarly Figure 2.
6. In Figure 3, the names of the compounds on the left are difficult to read. If possible, please increase the font size.
7. Lines 472 - 479. This paragraph should probably be included in the Introduction. Alternatively, you can leave it in Results, but you should shorten it a bit.
Author Response
Dear reviewer
Thank you for your valuable feedback. We have addressed all of your comments and suggestions in the manuscript.
1.A minor mistake. Now is: "2.2. Preparation of koji and miso" should be "2.1. Preparation of koji and miso".
Thank you very much for the feedback. This has been changed.
- The authors should explain what miso paste is in the methodology. Not every reader has encountered this regional product.
Thank you for this comment. We have added an explanation in the methods section.
- Chapter 2.5. Please comment it. Why did the authors choose this particular fiber and this particular chromatographic column for their research?
Dear reviewer, we decided to choose this particular fiber and chromatographic column following the methodology from this article (which we cite in the manuscript):
- Feng, T., Wu, Y., Zhang, Z., Song, S., Zhuang, H., Xu, Z., Yao, L,. Sun, M. Purification, identification, and sensory evaluation of kokumi peptides from agaricus bisporus mushroom. Foods 2019, 8(2), 1–12. https://doi.org/10.3390/foods8020043
- Lines 253 - 263. Starting with: "We selected two main cereal production..." and ending with: "... of miso fermentation as detailed in Table 1". This is information that relates to the methodology and is unnecessarily in this chapter.
Thanks for the feedback. At the beginning of the article we believe providing information makes it easier for readers to follow.
- In graphs B, C, D, E in Figure 1, the authors could add standard deviations if the graphs are based on average scores. Similarly Figure 2.
Thank you for your comment. As we mentioned in the manuscript, it is not a quantification, rather it is an identification.
- In Figure 3, the names of the compounds on the left are difficult to read. If possible, please increase the font size.
Thank you for this comment. We have addressed this in Figure 3.
- Lines 472 - 479. This paragraph should probably be included in the Introduction. Alternatively, you can leave it in Results, but you should shorten it a bit.
Thank you for the feedback, we have kept it in the results, however it is shorter than before.